

# Direct detection of condensed particulate polycyclic aromatic hydrocarbons on a molecular composition level at low pg m⁻³ mass concentrations via proton-transfer-reaction mass-spectrometry

Tobias Reinecke[1], Markus Leiminger[1], Andreas Klinger[1], Markus Müller[1]

[1]IONICON Analytik GmbH, Innsbruck, 6020, Austria

*Correspondence to*: Markus Müller (markus.mueller@ionicon.com)

**Abstract.** Particle condensed polycyclic aromatic hydrocarbons (PAHs) are a group of toxic organic compounds that are produced by incomplete combustion of organic material e.g. via biomass burning or traffic emissions. Even at low long-term exposure levels, such as 1 ng m⁻³ of benzo(a)pyrene, PAHs are recognized to be detrimental to human health. Therefore, a

quantitative characterization of PAHs at sub-ng m⁻³ levels is important to examine precise long-term exposure.

A new ultrasensitive generation of proton-transfer-reaction mass-spectrometry (PTR-MS) instruments coupled to the CHARON particle inlet is highly capable of quantitatively detecting this toxic class of compounds at a molecular composition level, while offering a high temporal resolution of < 1 min and sub-ng m⁻³ limits of detection. To demonstrate the capabilities of this new CHARON FUSION PTR-TOF 10k instrument, we present a thorough characterization of summertime ambient

condensed PAHs in Innsbruck, Austria. With a mass resolution of > 14 000 (m/Δm at full width half maximum) and unprecedented sensitivities of up to 40 cps ng⁻¹ m³, a series of 9 condensed PAHs of four ($C_{16}H_{10}$) to six aromatic rings ($C_{26}H_{16}$) are identified among a plethora of organic compounds in ambient organic aerosol. With unprecedented one-minute 3-σ limits of detection between 19 to 46 pg m⁻³, quantitative time-series of these PAHs of lowermost mass concentrations are determined. To understand the sources and processes associated with these condensed summertime PAHs in greater detail, a matrix

factorization including the ~ 4 000 ionic signals detected by the CHARON FUSION PTR-TOF 10k is performed, representing the vast majority of ambient organic aerosol. A total of 10 factors and corresponding time-series can be identified. Known tracer compounds like levoglucosan, pinonic acid or nicotine consequently allow the assignment to individual organic aerosol sources and physico-chemical processes. PAH emissions from traffic are found to be minor contributors during this summertime sampling period. The highest concentrations of PAHs are identified in a mixed aged oxygenated organic aerosol,

followed by a biomass-burning and a cigarette smoking organic aerosol.



**TOC Figure**

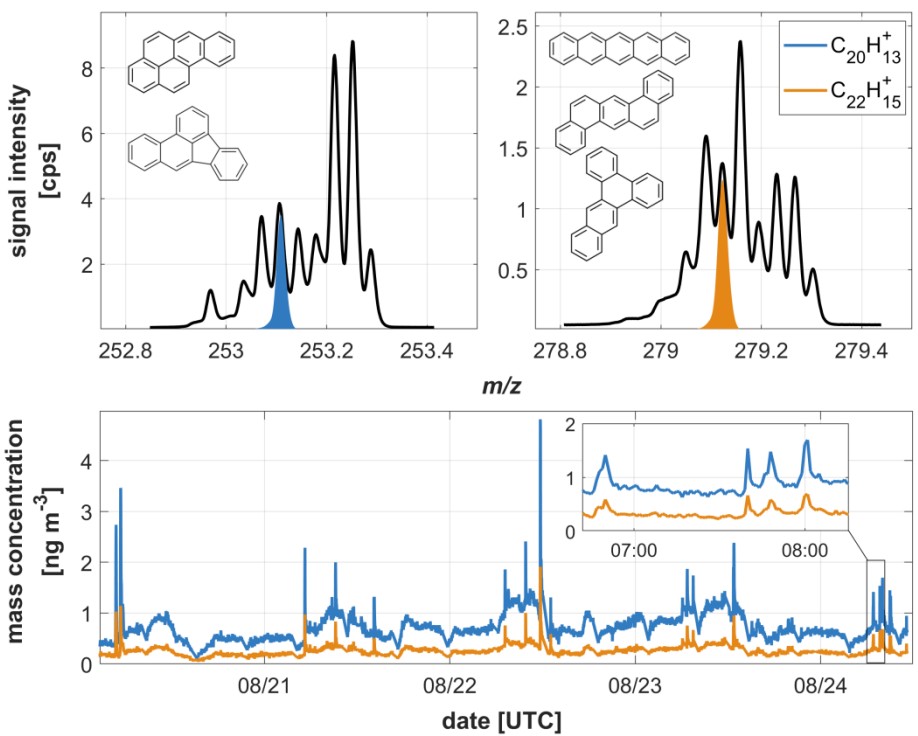



## 1 Introduction

Polycyclic aromatic hydrocarbons (PAHs) are a group of toxic organic compounds that are formed through incomplete combustion of organic materials. Common sources are biomass burning, industrial processes, transportation, and waste incineration (Kaur et al., 2013). PAHs are recognized as causing mutagenic, carcinogenic, teratogenic, and immunotoxicogenic effects (Agudelo-Castañeda et al., 2017). Even at low long-term exposure levels, such as 1 ng m$^{-3}$ of the commonly found benzo(a)pyrene, PAHs can be detrimental to human health (Choi et al., 2010). Once released, the semi-volatile nature of many PAHs, i.e. typically PAHs with more than three to four aromatic rings, leads to condensation onto ambient particles. Consequently, these particles are known to be widely spread via long range transport prior to deposition on soil, plants and water.

Although being a toxic compound class that is released by various sources, detecting PAHs can be challenging due to the semi-volatile nature and chemical properties of the most abundant PAHs. High throughput filter-based methods followed by desorption of the PAHs and analysis via HPLC or GC have proven to be very sensitive methods with low limits of detection (Thrane and Mikalsen, 1981; Borrás and Tortajada-Genaro, 2007; Lung and Liu, 2015). However, these methods are prone to sampling artefacts, e.g. via evaporation of more volatile PAHs while sampling (Patel et al., 2020). Sample handling, storage and analysis are elaborate and consume substantial resources. Additionally, filter measurements suffer from low time-resolution and therefore, source apportionment based on the data is not feasible.

In recent years, several direct methods based on mass-spectrometry have been developed (Laskin et al., 2018). The Aerosol Mass Spectrometer (AMS, Aerodyne Research Inc., USA) is probably the most commonly used instrument, allowing the characterization of total sub-µm particulate PAHs within single minutes and single-digit ng m$^{-3}$ mass concentrations (Dzepina et al., 2007; Poulain et al., 2011; Eriksson et al., 2014; Herring et al., 2015; Xu et al., 2022). However, due to interferences and fragmentation caused by electron ionization on a 600°C hot surface, the AMS is not able to provide data on individual PAH species. Other mass spectrometric techniques are based on laser desorption/ionization for a single particle based analysis. Such instruments like the REMPI-TOF give valuable insights including single-particle PAH-distributions in aerosols and additionally allow for an assignment of the detected particles to specific pollution sources (Passig et al., 2017, 2022). However, single-particle instruments do not provide mass concentrations of PAHs.

In contrast to these hard ionization techniques, soft chemical ionization mass spectrometry (CIMS) can conserve the chemical information and only exhibits a small amount of ionization induced fragmentation. While certain CIMS might be sensitive to derivatives of PAHs, with e.g. nitro or oxygenated functional groups that are often referred to as polycyclic aromatic compounds (PAC), most of these instruments cannot at all or only hardly directly detect and quantify PAHs since the chemical properties of PAHs, especially the low proton affinity, prevent an efficient ionization. One exception of a CIMS that is highly capable of detecting PAHs is proton-transfer-reaction mass-spectrometry (PTR-MS).

PTR-MS is a soft chemical ionization technique that can quantitatively detect a plethora of volatile organic compounds (VOCs) in real-time (Hansel et al., 1995; Graus et al., 2010; Yuan et al., 2017). It utilizes ion-molecule reactions of VOCs with $H_3O^+$



primary reagent ions in a well-controlled reaction environment. PTR-MS has already proven to be able to detect and quantify PAHs on a molecular composition level without ionization induced fragmentation (e.g. Gueneron et al., 2015). With the

CHARON particle inlet for PTR-MS (Eichler et al., 2015), this capability is extended to PAHs that are condensed onto particles (Müller et al., 2017; Piel et al., 2019). Side-by-side measurements of CHARON PTR-TOF and high throughput filter samples at the TROPOS operated research station in Melpitz, Germany, have already resulted in good qualitative and quantitative agreements on a 24 h basis (Wisthaler et al., 2020). However, CHARON PTR-TOF was able to provide significantly higher temporal resolution, even at single digit ng m$^{-3}$ mass concentrations.

For this study we have modified a new ultrasensitive PTR-MS instrument, the so-called FUSION PTR-TOF 10k (Reinecke et al., 2023), to be successfully paired to a further improved version of CHARON. To demonstrate the capabilities of this instrument to detect ultralow mass concentrations of a series of PAHs, a dataset was acquired in summertime in Innsbruck, Austria, during 11 consecutive days. Based on this dataset we show that CHARON FUSION PTR-TOF 10k achieves detection limits down to low double digit pg m$^{-3}$ mass concentrations for PAHs. To further understand the sources and processes associated with these PAHs, we apply matrix factorization to the entire recorded dataset including ~ 4 000 recorded ions,

representing the vast majority of organic compounds in ambient aerosol.

## 2 Methods

### 2.1 FUSION PTR-TOF 10k

The FUSION PTR-TOF 10k was recently introduced by Reinecke et al. (2023). In a nutshell, the FUSION PTR-TOF 10k is

an ultrahigh sensitivity PTR-MS instrument that reaches sensitivities of up to 80 cps pptV$^{-1}$ and limits of detection (LOD) in a sub-pptV range while simultaneously conserving the well-defined ion chemistry of conventional PTR-MS.

With the novel fast-SRI ion source $H_3O^+$ primary reagent ions are generated in a $H_2O$ plasma discharge at highest purity while neutral interferences like the hydroxyl radical (OH) are significantly reduced. These $H_3O^+$ primary reagent ions subsequently react with the VOCs of a sample gas inside the newly developed FUSION ion-molecule reactor to produce protonated VOCs

(VOC.H$^+$). Here, a stack of concentric ring electrodes generates a static longitudinal electric field superimposed by a focusing transversal radio frequency (RF) field, maximizing the ion transfer into the time-of-flight (TOF) mass spectrometer. The well-controlled ion-molecule reaction conditions enable a quantitative ionization of a wide range of VOCs, from nonpolar compound classes like aromatic species and PAHs to highly polar ones like highly oxidized organic molecules (HOMs). Finally, the ions are detected by TOF-MS with a mass resolution (m/Δm at full width half maximum; FWHM) in a range of

10 000 to 15 000. This mass resolution is sufficiently high to directly assign chemical compositions to the detected exact *m/z*. Herein the FUSION PTR-TOF 10k was operated at a reaction chamber temperature of 120°C and a moderate reduced electric field strength of E/N = 100 Td (1 Td = $10^{-17}$V cm$^2$; with E being the electrical field strength in the ion-molecule reactor and N being the number density of the sample gas the ion-molecule reactor). Together with the extended volatility range (EVR)

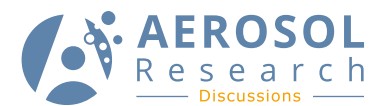

coating of both the inlet and the FUSION RF ion-molecule reactor, organic compounds of low volatility can be detected with

quick instrumental response times (Piel et al., 2021).

The original design of the FUSION PTR-TOF 10k reported by Reinecke et al. (2023) used an inlet flow of 100 - 120 ml min$^{-1}$. This flow is significantly higher than in standard PTR-MS, preventing a successful coupling to the aerodynamic lens system of the CHARON. Therefore, the FUSION RF ion-molecule reactor and inlet system were redesigned to accommodate a lower inlet flow rate in the range of 20 ml min$^{-1}$ and thus being compatible with the CHARON particle inlet.

## 2.2 CHARON Particle Inlet

With the "**ch**emical **a**nalysis of o**r**ganic particles **on**-line" (CHARON) particle inlet the capability of a PTR-TOF instrument to measure volatile organics is extended to the particle phase (Eichler et al., 2015; Müller et al., 2017). CHARON consists of a charcoal monolith denuder to remove the gas phase with an efficiency of > 99.999% for e.g. acetone. Simultaneously more than 90% of the particles above 70 nm are transmitted. These particles are then collimated by a high pressure aerodynamic

lens system operated at 7.5 mbar and a flow of ~450 ml/min. An enriched particle stream (~20 ml/min) is subsampled by means of a virtual impactor while the majority of residual gas is pumped away. Under the assumption that all particles are being subsampled, this setup allows for a theoretical particle enrichment of a factor 22.5. Finally, the volatile fraction of the particles is efficiently evaporated in a thermal desorption unit at reduced pressures (< 7.5 mbar) and moderate temperatures of 160°C. All volatilized organics are consequently detected in the gas-phase with a PTR-TOF instrument. Zero calibration of

the CHARON is achieved by redirecting the aerosol sample flow through a high-efficiency particulate absorbing (HEPA) filter.

With this setup, organics with a saturation mass concentration of log $C^0_{300 K}$ > -5, which includes parts of extremely low volatile organic compounds (ELVOC) plus the full range of low and semi volatile organic compounds (LVOC and SVOC, respectively), can be completely evaporated and detected in real-time. Figure S1 demonstrates this real-time response of the

CHARON FUSION PTR-TOF 10k for polydisperse levoglucosan particles (SVOC, log $C^0_{300 K}$ ~ 0) of roughly 0.5 µg m$^{-3}$ mass concentration. After switching to HEPA, an 1/e decay is reached within 8 s and a decay down to 10% within 31 s.

For this study, a computational fluid dynamics (CFD) optimized geometry of the aerodynamic lens system was tested that successfully increased the detectable particle size range by roughly 20 nm towards smaller particle sizes. The range of near-constant particle transmission (herein defined as ± 20%) now covers 100 nm up to > 1 µm. Particles in a size range from 60 to

100 nm are detected, but are observed with a reduced particle enrichment efficiency (see Figure S2 for the measured particle enrichment efficiency as a function of the particle size).

## 2.3 Site Description and Meteorological Conditions

The CHARON FUSION PTR-TOF 10k was deployed in Innsbruck, Austria, from August 18 to August 28, 2023. The measurements were conducted at the laboratory of IONICON Analytik, located in the East of Innsbruck. The meteorological

conditions in Innsbruck as an urban alpine environment are described in detail by Karl et al. (2020). The aerosol was sampled

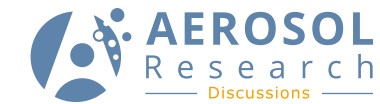

from the outside through a ¼" stainless steel tubing of < 2 m total length and a flow rate of 0.5 l min$^{-1}$. The sampling period mostly fell within a stable high-pressure period of low wind and elevated temperatures ranging from 17 to 34°C. A clear change in the weather pattern with high winds and lower temperatures was observed towards the end of the period. Figure 1, top panel displays the ambient temperature and the ozone mass concentration as measured in a close-by air quality site,
Innsbruck Reichenau - Andechsstraße, operated by the Austrian Umweltbundesamt (Environment Agency Austria; https://luft.umweltbundesamt.at/pub/gmap/start.html#). A planned power-line service in the afternoon of August 24[th] demanded an overnight interruption of the data acquisition.

**2.4 Data Acquisition**

All data was recorded with a 10 s time-resolution and mass spectra ranged up to *m/z* 719. To increase the separation capability
of isobaric ions, an upper-limit mass-resolution of the TOF-MS of > 14 000 (FWHM) was selected. We automatically conducted frequent zeros every 6 hours by switching to CHARON HEPA mode to remove the particle phase from the ambient air sample. A calibration with a dynamically diluted VOC standard was conducted prior to the measurement period to determine the transmission function of the instrument. The validity of this transmission function was checked at the end of the measurement period and was found to agree within a deviation of 10%. Sensitivities of the FUSION PTR-TOF 10k were in
the range of 15 to 20 cps pptV$^{-1}$ for most VOCs in the calibration gas standard. Together with the CHARON particle inlet this roughly corresponds to a sensitivity of ~ 40 cps ng$^{-1}$ m$^3$ for particulate PAHs.

**2.5 Data Analysis**

Data Analysis was performed with the IONICON Data Analyzer (IDA) in version 2.2.0.4 (Müller et al., 2013). IDA provides project management, high time- and *m/z*-resolved peak analysis and quantification of PTR-TOF datasets at highest precision
and accuracy. IDA's high level of automation and parallelization allows for fast analysis results even for complex datasets like CHARON particle spectra, which often include thousands of ionic signals.

To enable quantification of the dataset, the instrumental transmission function is combined with reaction rate constants (k-rate). The reaction rate constants of the compounds are calculated from the polarizability, obtained from a parameterization based on the respective chemical composition (Bosque and Sales, 2002), and the dipole moment, based on the parameterization
proposed by Sekimoto et al. (2017), by applying Su and Chesnavitch's parameterization of ion-polar molecule collisions (Su and Chesnavich, 1982). We estimate the combined accuracy of the transmission function and the parameterized k-rates in the range of ±30%. CHARON bulk information was corrected for fragmentation as introduced by Leglise et al. (2019). With this fragmentation correction, CHARON is typically able to quantify between 80 and 100% of the total organic aerosol.

In a first analysis step, monoaromatic and polyaromatic compounds are tentatively identified via the aromaticity equivalent
$X_C$ (Yassine et al., 2014). With the number (#) of C, N, H and O atoms and m as the fraction of the oxygen atoms involved in the **π**-bond structure for a given organic compound, $X_C$ is calculated according to Eq. (1):

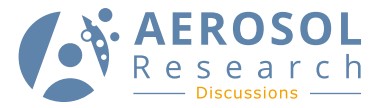

$$X_C = \frac{2\#C + \#N + \#H - 2m\#O}{RDBE - m\#O} + 1$$

with $RDBE = 1 + \frac{1}{2}(2\#C - \#H + \#N)$,                  (1)

Subsequently, pure hydrocarbons $C_xH_y$ with a ring and double bond equivalent (RDBE) ≥ 7 and with x > y are identified as

PAHs (with x being the number of C atoms and y the number of H atoms), as introduced for CHARON PTR-TOF by Piel et al. (2019).

To better understand the sources and processes associated with the detected signals, a matrix factorization was performed. This factorization is based on a nonnegative matrix factorization (NMF) with a nonnegative double singular value decomposition (NNDSVD) as described by Boutsidis and Gallopoulos (2008). This NNDSVD initialization approach leads to rapid reduction

of the approximation error of NMF and is therefore well suited for the factorization of large datasets like the one presented herein, that includes thousands of time series of ionic signals with each consisting of more than a hundred thousand data-points. In addition, similar to positive matrix factorization (PMF) that is frequently used in aerosol research (e.g. Ulbrich et al., 2009), this method results in a quantitative and qualitative reconstruction of time-series and mass spectral information. The optimal number of factors is selected via a cost function and a cross-correlation matrix of the corresponding time-series and

mass-spectra.

## 3 Results and Discussion

### 3.1 Identification and Quantification of PAHs

Figure 1 depicts the time-series of the ambient temperature and the ozone mass concentration, as published by the Austrian Umweltbundesamt, as well as the total signal of condensed organics and the average mass spectrum as recorded by CHARON

FUSION PTR-TOF 10k during the measurement period from August 18 to August 28, 2023. We again note that unfortunately the measurement had to be interrupted at around noon of August 24 due to a scheduled general service of the power lines in our laboratory and hence there is an 18 h gap in the recorded data.

Initially, in the period from 08/18 to 08/25, the total organics mass concentration slowly ramps up from below 1.5 µg m$^{-3}$ daily maximum to almost 5 µg m$^{-3}$, with highest mass concentrations in the noon hours. The visually clear correlation with ozone

mass concentrations indicates the significance of secondary particle formation processes during this period. Starting in the early morning of 08/27, the initial stable high pressure system gets replaced by considerably colder, stormy and wet conditions. As expected, with this weather change the particle concentration is significantly decreased with average mass concentrations below 1 µg m$^{-3}$. Despite this general trend, frequent short-term spikes especially during working days (i.e. 08/21 to 08/26, 08/28) show the presence of local particle emission sources with mass concentrations up to 11 µg m$^{-3}$.

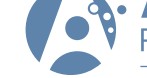



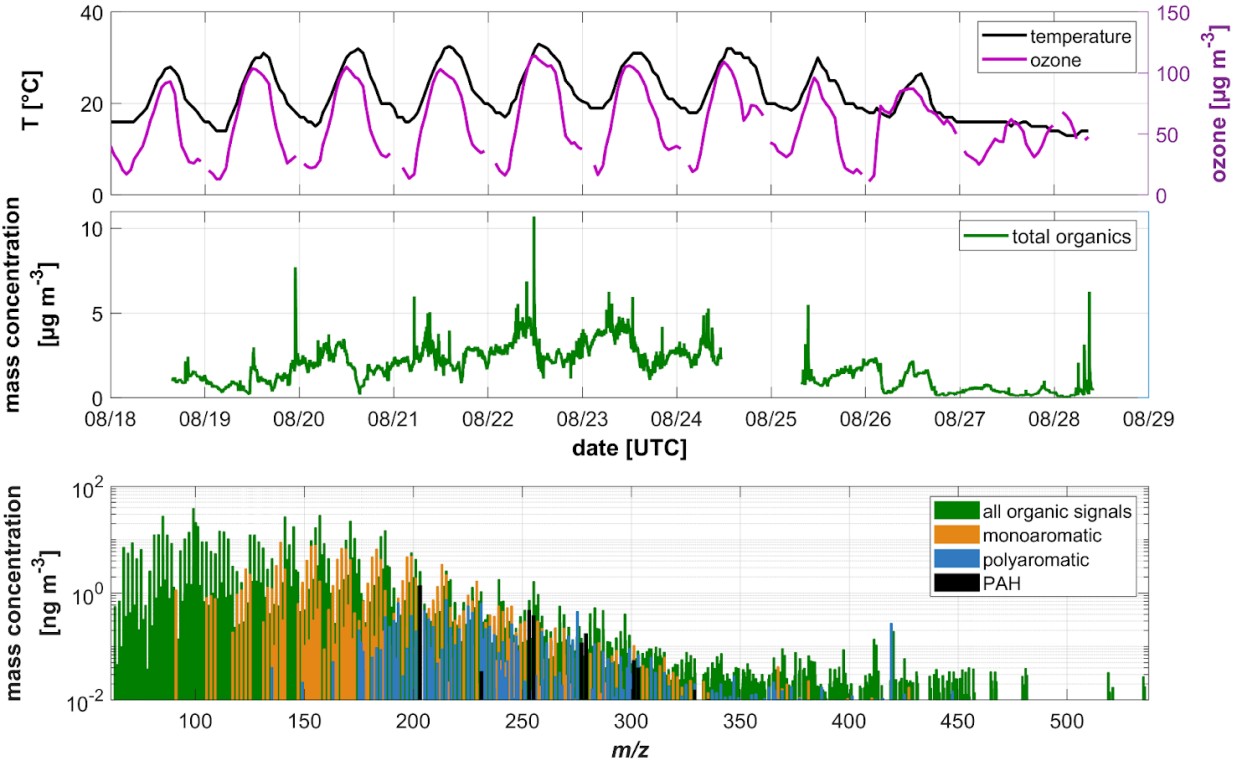


**Figure 1: Ambient temperature and ozone concentrations, published by the Austrian Umweltbundesamt (top panel), the mass concentration of total organics (middle panel) and the instrumental background corrected average mass spectrum (bottom panel) as recorded by FUSION PTR-TOF 10k. Color codes of the average mass spectrum reflect all detected organic signals (green), compounds that are tentatively identified as monoaromatics (orange; $2.5 \leq X_C < 2.71$, m = 0) or polyaromatics (blue; $X_C \geq 2.71$, m =**
**0) and the assigned PAHs (black; RDBE ≥ 7).**

The average mass spectrum (Figure 1, bottom panel) underlines the chemical complexity of the vaporized condensed organics. Repetitive groups of chemical compositions up to $m/z$ 350 are visible, representing compounds of various oxidation states. To further understand the chemical composition of the detected ionic signals, we have calculated the aromaticity equivalent $X_C$ to visualize an upper limit (m = 0) of monoaromatic ($2.5 \leq X_C < 2.71$) and polyaromatic ($X_C \geq 2.71$) signals. Black bars highlight

the detected PAHs that are $C_xH_y$ signals of RDBE ≥ 7 with x > y. In total, 9 PAH related ionic signals are identified that range from $C_{16}H_{11}^+$ ($m/z$ 203.086; four ring PAHs, e.g. pyrene, fluoranthene and other isomers) to $C_{26}H_{17}^+$ ($m/z$ 329.133; six-ring PAHs, e.g. hexacene). With the exception of $C_{16}H_{11}^+$, all PAH signals show an average mass concentration of well below 1 ng m$^{-3}$.



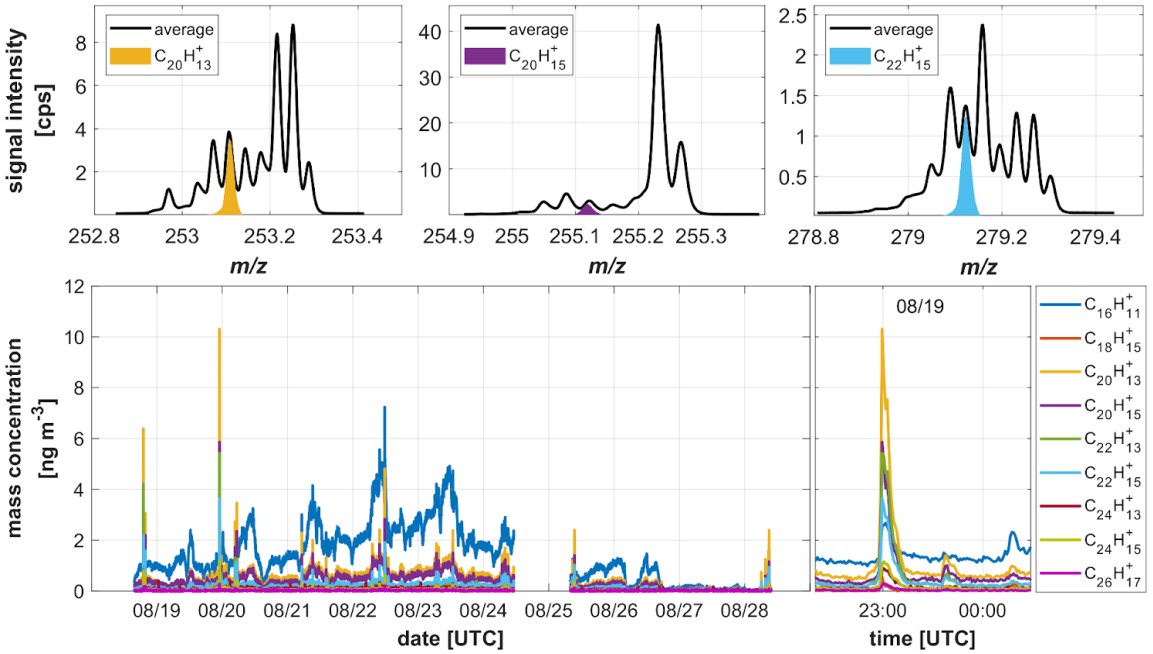

**Figure 2: Three exemplary peak systems of the PAH signals $C_{20}H_{13}^+$, $C_{20}H_{15}^+$ and $C_{22}H_{15}^+$ (top panels). Time traces of all 9 ionic signals from PAHs detected in the particle phase with a zoom into the evening hours of August 19, 2024 (bottom panels).**

Figure 2, top panel, exemplarily depicts three peak systems that include the PAH signals $C_{20}H_{13}^+$, $C_{20}H_{15}^+$ and $C_{22}H_{15}^+$. The complexity of these peak systems illustrates the importance of an instrument with high mass resolution to distinguish the PAH signals from the multitude of surrounding peaks. However, even at m/Δm > 14 000 (FWHM), peak separation remains

challenging. Nonetheless, a total of 9 ionic signals from PAHs detected in the particle phase (from $C_{16}H_{11}^+$ to $C_{26}H_{17}^+$) were extracted from our dataset. The time-series of these PAHs are plotted in the bottom panels of Figure 2. Highest average mass concentrations are visible for $C_{16}H_{11}^+$ (e.g. pyrene and isomers), whereas most spikes are dominated by $C_{18}H_{15}^+$, i.e. the sum of benzofluoranthenes and benzopyrenes. The lower right panel of Figure 2 shows a zoom into the late hours of 08/19, where the largest of all PAH spikes was recorded. This is a good example to assess the instrumental response time to our compounds

of interest in the experimental environment: all PAHs react quickly to this concentration increase and also drop quickly to previous background concentrations once the recorded plume has passed. In addition, this figure also indicates the extremely low noise levels of the recorded data. Even during this local emission event, most PAHs do not even exceed 1 ng m⁻³ levels. Based on the frequent HEPA measurements, the single minute 3-σ limit of detection was found to be between 19 and 46 pg m⁻³ for all detected PAH signals reported herein.

The high instrumental stability and separation capability, high time resolution, good response to temporal variations and extremely low limits of detection not only for the PAHs, but also for a wide range other organics, results in a good data quality that acts as an excellent basis for source apportionment.

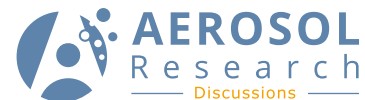

### 3.1 Source Apportionment

To obtain a both quantitative and qualitative reconstruction of the time-series based on mass spectral information assigned to
different factors, we performed a NMF with NNDSVD initialization. Subsequently, the sources or physico-chemical processes
are identified from the chemical information in the factor mass spectra (e.g. via well-known tracer compounds) and/or by
looking at the temporal or diurnal variations (e.g. to understand local emissions of the industrial area).

During the automated analysis of the dataset the number of factors, representing the sources and processes, was subsequently
increased. Using 10 separate factors for the NMF leads to a sufficient reduction of the cost function while the inclusion of
more factors did not further improve the accuracy of the reconstruction and, hence, did not add more chemical information.

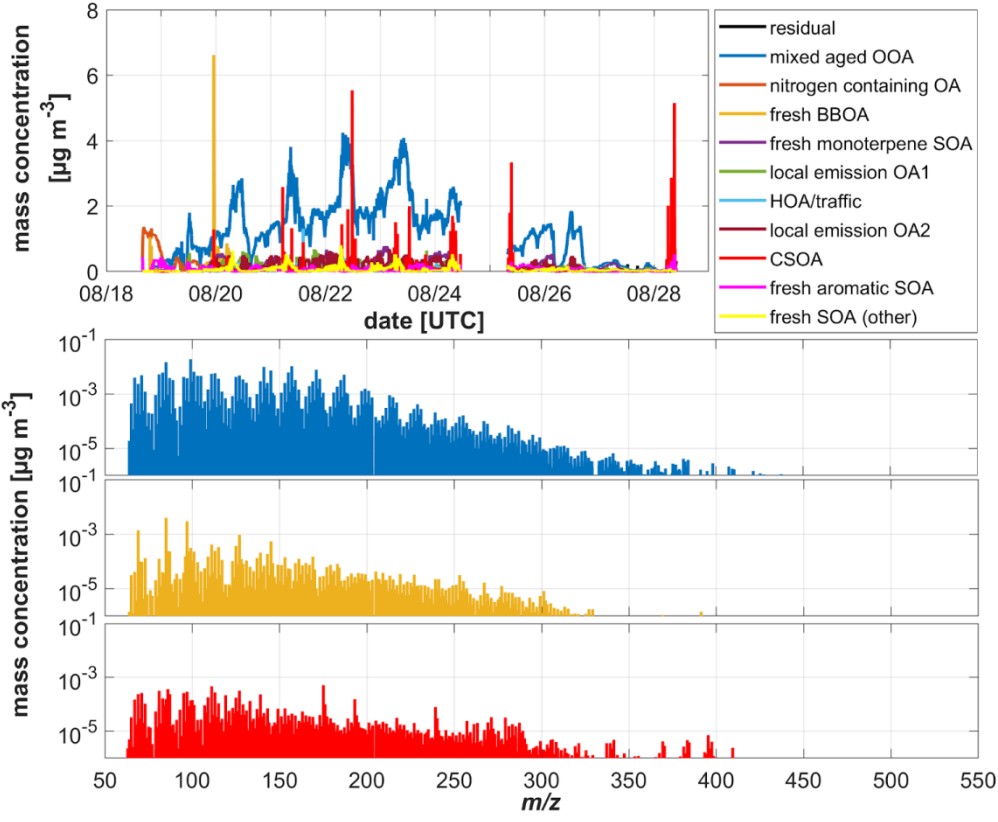

**Figure 3: Time traces of the 10 identified factors are shown in the top panel. Bottom panels depict the reconstructed mass spectra for a mixed aged oxygenated organic aerosol (OOA; blue), fresh biomass burning organic aerosol (BBOA; orange) and cigarette smoking organic aerosol (CSOA; red).**

The time series of these 10 identified factors are displayed in Figure 3, top panel. Note that the sum of the ten factors equals
the trace of total organics (with only a negligible residual). Figure 3, bottom panels, show the reconstructed mass spectra for a
mixed aged oxygenated organic aerosol (OOA), fresh biomass burning organic aerosol (BBOA) and cigarette smoking organic
aerosol (CSOA).

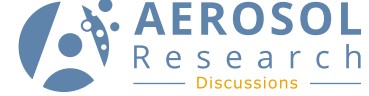

These three factors are selected as they include the majority of PAH related information. Find mass spectra (Figure S3) and
diurnal variations (Figure S4) of all ten factors in the supplement. A detailed description and interpretation of all factors
including their associated sources and processes lies outside the scope of this study.

The most dominant factor contains a plethora of compounds but is predominantly composed of mixed aged OOA with
compounds like levoglucosan and pinonic acid in different oxidation states. This factor generally shows highest mass
concentrations before noon, slightly drops in the afternoon, but stays dominant also during night. Our understanding is that
this factor mostly contributes to the accumulation of condensed organics during the stable weather period. Hence, this complex
factor could not be separated any further while introducing more NMF factors.

Another factor that shows two distinct spikes on the first weekend is attributed to fresh biomass burning. As can be seen in the
mass spectrum in Figure 3, middle panel, the factor contains the mass spectral signature of the anhydro-sugar levoglucosan
($m/z$ 85.028, 127.039, 145.050, 163.060), a well-known particulate tracer for biomass burning. Because these are singular
events on the first weekend with sunny and dry weather, the source could potentially be a nearby camp fire and barbecue.

During working hours, another prominent source of organic aerosol is cigarette smoke, most likely from smokers around the
building and on the building's balconies. Even in 2024, cigarette smoking is a widespread bad habit in the Austrian population
(Dorner et al., 2020). The mass spectrum of the respective factor (Figure 3, bottom panel) shows the expected compounds
found in cigarette smoke like nicotine and scopoletin (e.g. Arndt et al., 2020).

Following the separation of the total condensed organics into different factors and subsequent assignment of distinct sources
to each factor, we investigate in what way those sources contribute to the emission of condensed PAHs.

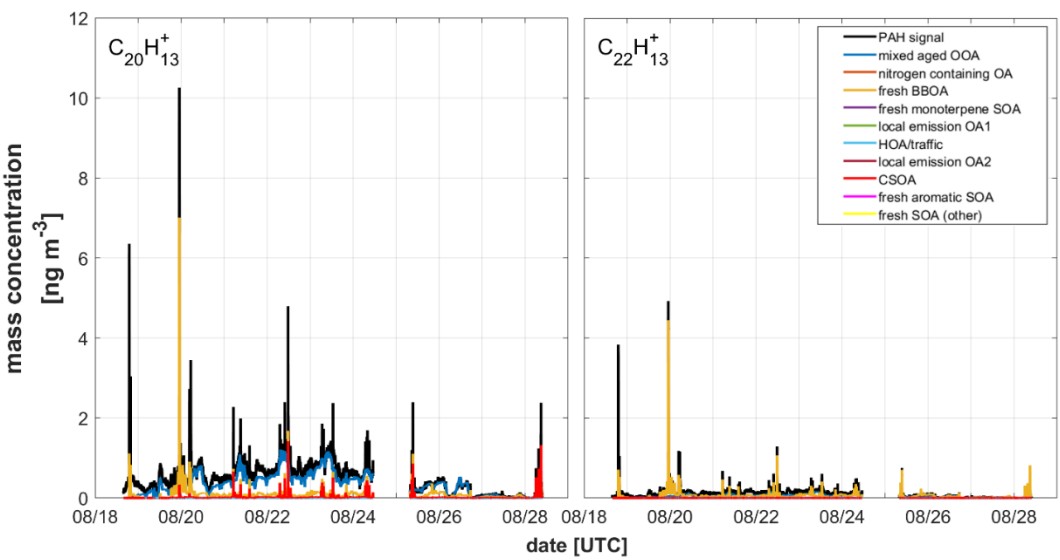

**Figure 4: Time series of two selected PAH signals ($C_{20}H_{13}^+$ (left) and $C_{22}H_{13}^+$ (right); black) that are selected to represent two separate emission groups. Colored traces show the individual NMF factors that contribute to the recorded signals.**



Figure 4 shows the time-series of two exemplary PAH signals (left: $C_{20}H_{13}^+$, right: $C_{22}H_{13}^+$) that are selected to represent two separate emission groups. Colored traces show the individual NMF factors that contribute to the recorded signals. Three factors, i.e. mixed aged OOA, fresh BBOA and CSOA, show the highest contribution to $C_{20}H_{13}^+$. On the other hand, $C_{22}H_{13}^+$ is dominated by the fresh BBOA emission factor. Obviously, fresh BBOA and CSOA coincide during times of cigarette smoking plumes, but still NMF is capable of separating the biomass burning fraction from a cigarette specific factor within

these short plumes, highlighting the general separation capability of this factorization method.

Also important to note is that traffic emissions only play a subordinate role in the processes associated with PAH emissions during the time of our observations.

## 4 Conclusion

We have successfully coupled a redesigned FUSION PTR-TOF 10k to a CHARON particle inlet with an improved

aerodynamic lens system for particle enrichment. With this instrument organic aerosol in Innsbruck, Austria, was analyzed over the course of 11 days in August 2023. The high sensitivity of the FUSION RF ion molecule reactor (S = 15 000 - 20 000 cps ppbV$^{-1}$ at an extended mass range up to $m/z$ 719) combined with the enrichment factor of 20 for particles in the size range from 100 nm to > 1 µm allowed the measurement of mass concentrations in the low double digit pg m$^{-3}$ range with one-minute time resolution. Furthermore, the identification capability of the high mass resolution TOF-MS (R > 14 000) enabled separating

the complex mass spectra into more than 4 000 ionic signals of organic aerosol. Among those thousands of signals, we were able to identify a series of 9 chemical compositions that represent a series of PAHs. Mass concentrations of these PAHs from 0 to 11 ng m$^{-3}$ are recorded; single minute 3-σ limits of detection were found to be between 19 and 46 pg m$^{-3}$.

Factorization of the entire dataset showed 10 separate sources and processes that affect organic aerosol concentration and composition. Out of these 10 identified factors three show significant contributions of PAHs: a mixed aged OOA factor, fresh

BBOA and CSOA. No significant contribution of PAHs could be identified in the traffic related factor.

All presented results and methods act as a proof-of-principle study for the 2024 ASIA-AQ mission (https://espo.nasa.gov/asia-aq), where a CHARON FUSION PTR-TOF 10k, identical in construction and performance, is installed aboard the NASA DC8 Airborne Laboratory. Goal of this three-month multinational mission is the holistic characterization of air pollutants, including condensed PAHs, with airborne, ground and satellite-based measurements in Southeast Asia.

## Data Availability


All data can be provided upon request by the corresponding author.



## Author Contributions

MM and TR conducted all hardware modifications to enable the coupling of FUSION and CHARON. AK designed the improved ADL and supported implementing the experimental setup. MM, ML and TR conducted the measurements, analyzed
the data and wrote the manuscript.

## Competing interests

TR, ML, AK and MM work for IONICON Analytik GmbH, which is commercializing CHARON and FUSION PTR-TOF 10k.

## Acknowledgement

We sincerely acknowledge the Austrian Research Promotion Agency (FFG) funding for the pSAT project (FO999900547) supporting IONICON's participation in NASA's ASIA-AQ campaign.

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
