# Peer review of "Direct detection of condensed particulate polycyclic aromatic hydrocarbons on a molecular composition level at low pg m-3 mass concentrations via proton-transfer-reaction mass-spectrometry"

_Aerosol Research, 2024_

## Referee Comment (RC1)

**Review of ar-2024-11**

The authors describe an application experiment in Innsbruck, Austria to demonstrate the capabilities of the CHARON inlet coupled to a high-resolution PTR-MS to detect PAHs with high sensitivity and time resolution in ambient air. The paper encompasses the concept of the instrument, some technical details, a matrix factorization data approach to deal with the high amount of data acquired by the instrument and its interpretation.

The study is interesting and sound and illustrates the impressive capabilities of the developed instrumentation. There are some points that could help to further improve the manuscript:

My main concern is that the paper sometimes sounds like an advertisement. This is not only inappropriate for a scientific paper but also unnecessary in view of the impressive results. Some examples: "…is highly capable…"(L12), "…highest precision and accuracy. IDA's high level of automation and parallelization allows for a fast analysis even for complex datasets."(L144), "The high instrumental stability and separation capability, high time resolution, good response to temporal variations and extremely low limits of detection […] good data quality […] excellent basis…" (L215). Etc. It is evident that this is a commercial instrument. However, potential customers will undoubtedly comprehend its benefits upon reading this paper, and they will not require such phrases and buzzwords.

Further comments:
- L16/17: repetitive use of "unprecedented"
- L17: could it be helpful to explain "3-σ limits"?
- L19: greater detail -> more detail
- L21: I have concerns with the phrase: "representing the vast majority of ambient organic aerosol." Can we know this? What means majority? Mass Concentration or number of species?
- L52: What is the "REMPI-TOF"? For single particles, there is the LDI approach (hard ionization) and a soft gas-phase method for PAHs, namely REMPI with prior laser desorption. The combination was published by Schade et al., Anal Chem. 2019, 91, 15, which replaced the former method by Passig et al., 2017. An application is correctly cited (Passig et al., 2022).
- L55: REMPI is a very soft ionization technology.
- L67: "TROPOS operated": Please be sure to include an explanation of any abbreviations used. (also REMPI).
- L76: repetition of L21.
- L82: SRI?
- L97: "preventing a successful coupling"? preventing→allowing for?
- L103: I do not understand the sentence beginning with "Simultaneously…"
- L 109: "All" → The…
- L117-121: Consider to provide more details in the SI. Increased size range compared to former Charon inlet? Shouldn't the ADL system be described before the vaporization?
- L125: It would be beneficial to consider a few words on the meteorological conditions, rather than merely referencing Karl et al.
- L134/135: "To increase the …" Was the mass resolution reduced compared to the instruments full mass resolution?
- L162/163: Repetitions. Please rephrase.
- L175-177: Not necessary to mention the reason for the blackout. No one doubts the stability of the instrument.
- L234-236: Some language editing might be helpful.
- Figure 4: I recommend that the plots are not displayed next to each other, but on top of each other. This would make it much easier to compare the timelines and to see the coincidence of the plume events.

---

## Author Response (AR1)

**Response to Reviewers' Comments for Manuscript AR-2024-11**

Dear Editor and Reviewers,

Thank you for your thoughtful and constructive comments on our manuscript entitled "Direct detection of condensed particulate polycyclic aromatic hydrocarbons on a molecular composition level at low pg m$^{-3}$ mass concentrations via proton-transfer-reaction mass-spectrometry". We appreciate the time and effort you took to review our work and provide feedback. We have carefully considered each comment and have made several revisions to our manuscript accordingly. Below, we address each point raised and describe the amendments we have made to enhance the quality and clarity of our manuscript.

**Response to Referee Comment 1**

*The authors describe an application experiment in Innsbruck, Austria to demonstrate the capabilities of the CHARON inlet coupled to a high-resolution PTR-MS to detect PAHs with high sensitivity and time resolution in ambient air. The paper encompasses the concept of the instrument, some technical details, a matrix factorization data approach to deal with the high amount of data acquired by the instrument and its interpretation.*

*The study is interesting and sound and illustrates the impressive capabilities of the developed instrumentation.*

*There are some points that could help to further improve the manuscript:*

*My main concern is that the paper sometimes sounds like an advertisement. This is not only inappropriate for a scientific paper but also unnecessary in view of the impressive results. Some examples: "…is highly capable…"(L12), "…highest precision and accuracy. IDA's high level of automation and parallelization allows for a fast analysis even for complex datasets."(L144), "The high instrumental stability and separation capability, high time resolution, good response to temporal variations and extremely low limits of detection […] good data quality […] excellent basis…" (L215). Etc. It is evident that this is a commercial instrument. However, potential customers will undoubtedly comprehend its benefits upon reading this paper, and they will not require such phrases and buzzwords.*

We have revised the manuscript to ensure it adheres to a more academic tone, focusing on presenting our results and methodology without promotional language.

*Further comments*

*- L16/17: repetitive use of "unprecedented"*

We have removed both instances of unprecedented (to further reduce promotional language).

*- L17: could it be helpful to explain "3-σ limits"? - L19: greater detail -> more detail*

We have removed the "3-σ" from the "3-σ limits of detection". We think this might be too much information for an abstract. "3-σ limits of detection" are now explained in the Results and Discussion section:

*"Based on the frequent HEPA measurements, the single minute limits of detection are derived based on the 3-σ variation of the recorded HEPA background signals. Hence, 3-σ limits of detection were found to be between 19 and 46 pg m⁻³ for all detected PAH signals reported herein."*

*- L21: I have concerns with the phrase: "representing the vast majority of ambient organic aerosol." Can we know this? What means majority? Mass Concentration or number of species?*

Indeed, this phrase might be misleading. In recent publications CHARON PTR-MS has shown to be able to measure the vast majority of mass concentration present in ambient air or atmospheric simulation studies (via intercomparison with other analytical instruments like SMPS, TOF-AMS, etc.). Number of species is more tricky based on two limitations: (a) CHARON PTR-MS is only able to detect species based on chemical compositions. Hence the number of actual species including isomers will be significantly higher than the reported number. (b) CHARON PTR-MS has a large but also limited volatility range with lowermost saturation mass concentrations that can be detected in real-time starting from log $C^0$ = -5 (i.e. "ELVOCs"). Species of lower volatility will not be completely evaporated or will not at all be evaporated. Of course, in ambient air we can not rule out the presence of such extremely low volatile species.

However, we believe referring to *"a vast majority of mass concentration of ambient organic aerosol"* is the most comprehensive option within the abstract.

*- L52: What is the "REMPI-TOF"? For single particles, there is the LDI approach (hard ionization) and a soft gas-phase method for PAHs, namely REMPI with prior laser desorption. The combination was published by Schade et al., Anal Chem. 2019, 91, 15, which replaced the former method by Passig et al., 2017. An application is correctly cited (Passig et al., 2022).*

The authors want to thank the referee for this clarification. We have adapted the original text to the following:

*"Other mass spectrometric techniques are based on laser desorption/ionization for a single particle based analysis. One instrument featuring a highly selective and soft laser ionization of PAHs is the resonance enhanced multiphoton ionization (REMPI) TOF (Schade et al., 2019, Passig et al., 2022). This REMPI-TOF gives valuable insights including single-particle PAH-distributions in aerosols and additionally allows for an assignment of the detected particles to specific pollution sources. However, single-particle instruments do not provide mass concentrations of PAHs."*

*- L55: REMPI is a very soft ionization technology.*

Indeed, hard ionization was initially referring to the electron ionization of the AMS. We rephrase this sentence to the following:

*"In contrast to these either hard or highly selective ionization techniques, soft chemical ionization mass spectrometry…"*

*- L67: "TROPOS operated": Please be sure to include an explanation of any abbreviations used. (also REMPI).*

We now introduce TROPOS via the following:

*"Leibniz Institute for Tropospheric Research (TROPOS; Leipzig, Germany)"*

and REMPI via:

*"resonance enhanced multiphoton ionization (REMPI) TOF"*

*- L76: repetition of L21.*

Fixed

*- L82: SRI?*

Fast-SRI ion source is now properly introduced and referenced via the following:

*"fast-selective-reagent-ion (SRI) ion source (see Reinecke et al., 2023, for details)"*

*- L97: "preventing a successful coupling"? preventing→allowing for?*

Preventing is correct. However we rephrased it to the following for more clarity:

*"This flow is significantly higher than in standard PTR-MS that is in the range of 15-25 ml min$^{-1}$ and exceeds the capability of the aerodynamic lens system of the CHARON particle inlet."*

*- L103: I do not understand the sentence beginning with "Simultaneously…"*

For more clarity we have modified this sentence:

*"Simultaneously, more than 90% of the particles above 70 nm are transmitted through the charcoal monolith denuder."*

*- L 109: "All" → The…*

Fixed

*- L117-121: Consider to provide more details in the SI. Increased size range compared to former Charon inlet? Shouldn't the ADL system be described before the vaporization?*

Indeed, even for the optimizations, it makes sense to consider the operational principles of CHARON. Hence we have moved the ADL improvements one section up. We also adjusted the CHARON size-limit description the following way for more clarity:

*"For this study, a computational fluid dynamics (CFD) optimized geometry of the aerodynamic lens system was tested that successfully increased the detectable particle size range by roughly 20 nm towards smaller particle sizes compared to earlier versions of CHARON. The range of near-constant particle transmission (herein defined as ± 20%) now covers ~100 nm up to > 1 µm (instead of ~120 nm up to > 1µm). Particles in a size range from 60 to 100 nm are detected, but are observed with a reduced particle enrichment efficiency (see Figure S1 for the measured particle enrichment efficiency as a function of the particle size)."*

*- L125: It would be beneficial to consider a few words on the meteorological conditions, rather than merely referencing Karl et al.*

Indeed, the Karl et al. (2020) reference feels a bit misplaced. However, we think the reference includes valuable information about the general meteorological conditions in Innsbruck. We have decided to move this information 3 sentences down, directly after the description of the meteorological conditions during the sampling site. We have added a short description about the environment of the measurement site for a better understanding of the role of the weather conditions.

*"The general meteorological conditions in Innsbruck as an urban alpine environment are described in detail by Karl et al. (2020). In brief, Innsbruck is a major city located in the river Inn valley at 570 m above sea level which is oriented from west to east and is surrounded by mountains of up to 2500 m in the north and south constraining air mass exchange. Local emissions by a typical mixture of urban, industrial and nearby agricultural sources are complemented by biogenic emissions from forests, which cover the mountain slopes, and transit traffic emissions, as the river Inn valley acts as the major transit route connecting Germany and Italy."*

*- L134/135: "To increase the …" Was the mass resolution reduced compared to the instruments full mass resolution?*

In the applied instrument mass resolution and sensitivity (generally spoken) follow opposite functions. The instrument presented herein is rated for 40 000 cps/ppbV of sensitivity for trimethylbenzene at a mass resolution of (more than) 10 000. For this application (and after running some preliminary, not reported, tests) we decided that a mass resolution of 10 000 is not sufficient to separate some of the compounds of interest. We have therefore sacrificed some of the instrument's sensitivity in favor of mass resolution.

To add the relevant information to the manuscript, we have added the following:

*"To increase the separation capability of isobaric ions, an enhanced upper-limit mass-resolution of the TOF-MS of > 14 000 (FWHM) was selected (instead of the rated mass-resolution of 10 000)."* and *"Sensitivities of the FUSION PTR-TOF 10k operated at mass resolutions of 14 000 were in the range of 15 to 20 cps pptV$^{-1}$ for most VOCs in the calibration gas standard."*

*- L162/163: Repetitions. Please rephrase.*

It was rephrased to the following:

*"To better understand the sources and processes associated with the detected signals, a nonnegative matrix factorization (NMF) was performed. This NMF is initialized via a nonnegative double singular value decomposition (NNDSVD) as described by Boutsidis and Gallopoulos (2008)."*

*- L175-177: Not necessary to mention the reason for the blackout. No one doubts the stability of the instrument.*

We have removed this information.

*- L234-236: Some language editing might be helpful.*

The respective lines were rephrased:

*"These three factors are selected as they include the majority of PAH related information. The mass spectra (Figure S4) and diurnal variations (Figure S5) of all ten identified factors are presented in the supplement. A detailed description and interpretation of all factors including their associated sources and processes lie outside the scope of this study."*

*- Figure 4: I recommend that the plots are not displayed next to each other, but on top of each other. This would make it much easier to compare the timelines and to see the coincidence of the plume events.*

The authors agree that this is a very good idea. Figure 4 was adjusted accordingly:

[Figure]

**Response to Referee Comment 2**

*General comments*

*The manuscript proposed by Tobias Reinecke et al. entitled "Direct detection of condensed particulate polycyclic aromatic hydrocarbons on a molecular composition level at low pg m-3 mass concentrations via proton-transfer-reaction mass-spectrometry" aims at presenting the*

*online detection of PAHs in atmospheric aerosol at very low concentrations using high resolution chemical ionization mass spectrometer. The manuscript is well constructed and very pleasant to read. The study provide sounds results from a simple field campaign conducted in Austria, showing successful detection of PAHs at environmental levels. However the paper sometimes lacks precisions, and more information should be provided to fulfil the requirement for publication in an peer reviewed international journal. I think that the paper contains valuable information that may be published after the authors improved it. My comments are listed below.*

*Specific Comments*

*Methods: It is written that ADL, IMR and inlet were modified but no details are provided. For sure it is not necessary to give all details, but a few words should be said to one can understand what has been done and evaluate how these changes can affect the detection of organic compounds from either gas or particle phase.*

Indeed, the IMR was adjusted for lower inlet flow (~20 instead of 100 ml/min; as written in the text). Regarding the aerodynamic lens system, referee 1 has also raised a similar question. Hence, we have added additional information as follows:

*"For this study, a computational fluid dynamics (CFD) optimized geometry of the aerodynamic lens system was tested that successfully increased the detectable particle size range by roughly 20 nm towards smaller particle sizes compared to earlier versions of CHARON. The range of near-constant particle transmission (herein defined as ± 20%) now covers ~100 nm up to > 1 µm (instead of ~120 nm up to > 1µm). Particles in a size range from 60 to 100 nm are still detected, but are observed with a reduced particle enrichment efficiency (see Figure S1 for the measured particle enrichment efficiency as a function of the particle size)."*

*L. 15 (also L135): at which m/z is the resolution 14 000 calculated?*

10 000 mass resolution at FWHM is roughly reached with m/z 45, 12 000 with m/z 65 and 14 000 with m/z 200. This means all PAH related signals reported herein are recorded with a mass resolution of approximately 14 000 or higher. The maximum recorded mass resolution for m/z > 330 is in the range of 14 500.

Although this is undoubtedly interesting information for some skilled readers, the authors prefer to stick to *"... an upper-limit mass-resolution of the TOF-MS of > 14 000 (FWHM)..."* for easy readability.

*L. 16-17 and in the manuscript: I understand the authors want to point how good is the new CHARON FUSION PTR-TOF 10k but the use of "unprecedent" twice in 2 lines is too much (and then through the paper, eg L.21 21 "greater detail"). The purpose is to show the detection of the PAHs and show the capability of the new instrument, which does not require the use many words like "unprecedent" and so on. Please revise the manuscript considering this.*

As already proposed by Referee 1, we have revised the manuscript to ensure it adheres to a more academic tone, focusing on presenting our results and methodology without promotional language. This also includes the elimination of buzz-words like unprecedented.

*L. 45: not true, see for example Srivastava et al. 2018 (STOTEN)*

Indeed, Srivastava et al. (2018) demonstrate the application of matrix factorization for source apportionment based on filter sampled. According to their publication, filters were taken on a daily basis. Srivastava et al. (2018) mention in their conclusion that a higher time-resolution (6h instead of 24h) would be helpful to separate traces of different diurnal variation. But even at 6h time-resolution, short term variations that only last for a few minutes as reported herein will be very difficult to be captured via matrix factorization.

However, to cope with this information, we have modified this sentence accordingly:

*"Additionally, filter measurements suffer from low time-resolution and therefore, source apportionment based on the data might miss important factors caused by diurnal or even quicker features."*

*Introduction: FIGAERO-CIMS technology or EESI orbitrap were not mentioned while they are other technique, which, depending on CI, might detect PAHs at a time resolution lower than typically done with traditional offline filter approach. These technics should be at least mentioned so that the introduction is complete. For instance it is not complete.*

Other methods are limited to instrumentation that have been reported to be able to detect PAHs. FIGAERO-CIMS, typically operated in iodide or bromide mode, is not able to detect PAHs. A water-cluster ionization mode is also possible, but proton affinities of PAHs are low and, if detectable at all, sensitivities might be highly reduced. EESI is typically operated in sodium cluster ionization mode. Also this reagent ion is, to our knowledge, not able to ionize PAHs. However, both techniques might be able to detect polycyclic aromatic compounds (PAC), i.e. PAH with one (or many) functional group(s). But as those

compounds are outside the scope of this manuscript, the authors decide not to add EESI and FIGAERO to the Introduction.

Please note, following text is already part of the introduction:

*"While certain CIMS might be sensitive to derivatives of PAHs, with e.g. nitro or oxygenated functional groups that are often referred to as polycyclic aromatic compounds (PAC), most of these instruments cannot at all or only hardly directly detect and quantify PAHs since the chemical properties of PAHs, especially the low proton affinity, prevent an efficient ionization."*

*L. 114-116 – Figure S1: The time response of the CHARON FUSION PTR-TOF is shown on figure S1 and shows that 10 % of the signal is still there after 31 s. What would be interesting instead is showing the time required to reach background. In other words what is shown here is that measurement at time resolution lower (i.e. faster) than 30 s is not possible due to the time response of the instrument. Was the time response tested for other compounds (especially heavier? More oxygenated ? PAHs?)*

Indeed, levoglucosan (a low volatile compound with a saturation mass concentration $logC^0$ of around 0) shows a decay to less than 10% within roughly 30 s. Quicker variations will be captured, but will be smoothed out. A drop from 10% signal intensity to background levels depends on the initial signal of the respective compound (as it follows an exponential decay and background concentrations might be compound dependent). Hence no absolute numbers can be given. Extremely low volatile compounds will show decay times to 10% even slower than 30 s. However, the herein reported 1 min time-resolved data will very likely capture the 1/e variations of even such extremely low volatile compounds. The response is clearly sufficient to capture the variations of condensed PAHs as shown in Figure 2 or the TOC figure for example.

A detailed overview of typical decay times of PTR-MS with extended volatility range (EVR) technology covering a wide range of compounds was published by Piel et al. (2021; as cited in the manuscript).

*L. 117-118: the modifications made to the ADL should be explained*

See changes above.

*L. 119-121: Was the particle enrichment factor estimated for other type of aerosols or organic compound? Just checking it is not affected by chemical composition (should not)*

Similar to other ADL systems, the size cut-off is dependent on particle size, density and shape. See e.g. Eichler et al. (2015) for more information. Levoglucosan is the compound of our choice as it is not toxic, easy to dissolve and nebulize, it does not evaporate in the DMA and, most importantly, levoglucosan is an important biomass burning marker in ambient aerosol.

*Figure 2: Is the peak system from the average spectra of the entire campaign? I suggest moving these mass spectra of peak systems to SI, it is not essential for the paper, as anyone knowing what a resolution of 14 000 (at which m/z?) means is convinced that most isobars can be separated.*

Indeed, the spectra labeled "average" are average spectra of the entire campaign. We have adjusted the caption of Figure 2 accordingly:

*"Figure 2: Three exemplary peak systems of the PAH signals $C_{20}H_{13}^+$, $C_{20}H_{15}^+$ and $C_{22}H_{15}^+$ (campaign average, top panels). Time traces of all 9 ionic signals from PAHs detected in the particle phase with a zoom into the evening hours of August 19, 2024 (bottom panels)."*

Presenting a mass resolution of more than 14 000 for signals > *m/z* 200 is key to separate the plethora of isobaric interferences. We therefore consider this important information and would prefer to keep it in the main manuscript.

*Placing the 9 PAHs on the same figure makes it difficult to read, especially for the ones with low concentrations. I suggest splitting the figure or use log scale.*

We tried the visualization of figure 2 in log scale. However, the crucial information that we want to highlight in this figure seems to get lost in log scale. We therefore decided to keep a linear y-axis.

*L. 209:211: It is not clear from figure 2 the time response is "quick" as the authors wrote. It took ca. 20 minutes for signal to reach back a low concentration after the peak observed the 09/18 at 23:00, which can reflect the real decay of the plume in the atmosphere, or this plume event might only be few minutes, and the rest is due to memory effect. If the authors do not have an external measurement (NOx for example) to confirm the observed trend on figure 2, I am not sure the authors can claim a real fast time response.*

The analysis of pure compounds like levoglucosan (as presented in the supplement) or others (as shown by Piel et al. 2021; cited) show the single-minute response capability of the instrument. The dynamics of ambient data certainly does not. However, the herein

claimed single-minute response of CHARON PTR-MS was already frequently reported, e.g. in comparison to CO during two separate airborne missions (Piel et al., 2019, and Pagonis et al., 2021). Hence, with all the reported data and clear agreement of e.g. the levoglucosan response, the authors are highly confident to claim a real fast time response.

*L. 215-217: these lines re explain the instrument capability, which was quite clear based on previous section. These lines are not necessary.*

These lines were adopted to scope with the comments of Referee 1.

*L. 224-225: the choice of 10 factors solution should be supported by a figure showing that adding more factors do not better explain variance. Therefore, examples of tracers used to identify the sources should be included.*

The authors are of the opinion, that lines 224 to 225 do very well explain the reasons for 10 factors:

*"Using 10 separate factors for the NMF leads to a sufficient reduction of the cost function while the inclusion of more factors did not further improve the accuracy of the reconstruction and, hence, did not add more chemical information."*

Hence, adding more factors does not further reduce the cost function but at the same time induces strong correlations of individual factorized time-series and mass-spectra. As this is a statistical analysis, there are no examples of individual tracers.

*L. 231: I suggest adding the sum of 10 factors and residual to SI.*

Indeed, we have added the figure to the SI and have added following reference to the text:

[Figure]

*"Figures S3: Visualization of the total measured organic aerosol and the respective NMF residual (i.e. the total measured organic aerosol minus the sum of all ten identified NMF factors)."*

*"The time series of these 10 identified factors are displayed in Figure 3, top panel. Note that the sum of the ten factors equals the trace of total organics (with only a negligible residual, see Figure S3 for more detail)."*

*L. 237-241: Naming this aged OOA is questioning as it contains significant amounts of PAHS, that are reactive compounds (Atkinson and Arey, 2007). They should be low in this factor if it corresponds to aged air mass. Wind speed and direction would help to confirm this is an aged OA factor.*

This factor is dominated by an aged OOA signal, but also mixed with other compounds like PAHs. Hence we abbreviated it to mixed aged OOA, as described in the manuscript. HySplit trajectory simulations were performed prior submission of this manuscript. Although the spatial resolution of HySplit clearly fails in an alpine environment like in Innsbruck, it indicates that the air masses are only shifted from east to west (and backwards) within the Inn valley during this period of particle build-up and aging.

*Figure 4: it is very difficult to see something on the right panel. Maybe a log scale can help better visualize concentration variations*

Referee 1 proposed a new orientation of the two figures that was incorporated into our manuscript.

*L. 257: Please be more precise and give numbers for the contribution of these factors to the PAHS.*

We have added the percentage of total PAH mass concentrations that contribute to these individual factors.

*"…a mixed aged OOA factor, fresh BBOA and CSOA, contributing to 63%, 19% and 6% of the total PAH signal, respectively. No significant contribution of PAHs could be identified in the traffic related factor (0.4%)."*

*Conclusion: Finally, out of 4000 ions, only 9 were analysed as PAHs. Are potentially other PAHS present? Or PAHs are a very minor contributor to summer OA in Innsbruck?*

To our knowledge, the 9 reported chemical compositions found in the condensed phase that correspond to PAHs are covering all the expected range. The lower panel in Figure 1 highlights the presence of other polyaromatic compounds. Summertime mass concentrations of organic aerosol is in general rather low in Innsbruck. The concentrations reported herein very likely represent a period of higher than usual concentrations. An obvious exception might be rare instances of close by wildfires.

*L. 276-279: rather than ending on a future campaign of the research group, which is not appropriated for research paper in an international peer reviewed journal, it would be interesting to conclude on the possibilities offered by the new instrument, and the potential new scientific findings or field that are opened with the capability of the new instrument.*

We agree. The previously "future" campaign is already a successful "past" campaign. Hence this section was deleted and replaced by the following:

*"All presented results and methods act as a proof-of-principle study that demonstrates the unprecedented analytical performance of such a CHARON FUSION PTR-TOF 10k. However, due to the complexity of the recorded data, the interpretation of the vast majority of signals and associated factors exceed the scope of this manuscript. Hence, interesting trends, including repetitive short term spikes, diurnal variations and also the impact of changing weather conditions, affecting the chemical composition of organic aerosol could not be studied in detail.*

*Future work will include an exploration of other, even softer, ionization techniques like soft ammonium adduct ionization (A.NH4+) that has been reported to conserve the chemical composition of a plethora of oxygenated organic compounds (e.g. Müller et al., 2020, Reinecke et al., 2023). This conservation of chemical information together with a high*

*selectivity to oxygenated organic compounds will allow for gaining an even deeper insight in primary emission and secondary formation processes of particulate oxygenated organic species in the atmosphere."*

*Minor Comments*

*L.134: I was curious to know why the upper m/z limit was so specific (719) ?*

In time of flight mass spectrometry you typically do set a pulsing frequency that defines a maximum time of flight for individual extractions. We put this to an even number and we were aiming to cover a mass range up to ~700. This resulted in an upper m/z limit of 719. As visible in the data, anything above m/z 500 would have been sufficient for this set of data.

*L. 162: please specify the software or code language used for source apportionment (NMF etc.)*

This functionality is also integrated in the data analysis software IDA. As we already state that IDA in version 2.2.0.4 was used for data analysis in the first sentence of this section we refrain from reciting IDA again.

*Please make sure abbreviations are all explained*

We tried to explain all abbreviations in scope of Referee 1.

In conclusion, we believe these revisions have addressed the concerns raised by the reviewers and have significantly improved the manuscript. We appreciate the opportunity to revise our work and hope that the changes meet the journal's standards for publication.

Thank you for considering our manuscript for publication in the Journal of Aerosol Research. We look forward to your feedback and hope for a positive decision.

Sincerely,

Markus Müller

**References**

Eichler, P., Müller, M., D'Anna, B., and Wisthaler, A.: A novel inlet system for online chemical analysis of semi-volatile submicron particulate matter, Atmospheric Measurement Techniques, 8, 1353–1360, https://doi.org/10.5194/amt-8-1353-2015, 2015.

Pagonis, D., Campuzano-Jost, P., Guo, H., Day, D. A., Schueneman, M. K., Brown, W. L., Nault, B. A., Stark, H., Siemens, K., Laskin, A., Piel, F., Tomsche, L., Wisthaler, A., Coggon, M. M., Gkatzelis, G. I., Halliday, H. S., Krechmer, J. E., Moore, R. H., Thomson, D. S., Warneke, C., Wiggins, E. B., and Jimenez, J. L.: Airborne extractive electrospray mass spectrometry measurements of the chemical composition of organic aerosol, Atmospheric Measurement Techniques, 14, 1545–1559, https://doi.org/10.5194/amt-14-1545-2021, 2021.

Piel, F., Müller, M., Mikoviny, T., Pusede, S. E., and Wisthaler, A.: Airborne measurements of particulate organic matter by PTR-MS: a pilot study, Atmospheric Measurement Techniques Discussions, 1–20, https://doi.org/10.5194/amt-2019-181, 2019.

Schade, J., Passig, J., Irsig, R., Ehlert, S., Sklorz, M., Adam, T., Li, C., Rudich, Y., and Zimmermann, R.: Spatially Shaped Laser Pulses for the Simultaneous Detection of Polycyclic Aromatic Hydrocarbons as well as Positive and Negative Inorganic Ions in Single Particle Mass Spectrometry, Anal. Chem., 91, 10282–10288, https://doi.org/10.1021/acs.analchem.9b02477, 2019.